# Assessment of the functional efficacy of root canal treatment with high-frequency waves in rats

Saori Matsui[1,2], Naomichi Yoneda[3], Hazuki Maezono[1]*, Katsutaka Kuremoto[1], Takuya Ishimoto[4], Takayoshi Nakano[4], Hiromichi Yumoto[5], Shigeyuki Ebisu[1], Yuichiro Noiri[6], Mikako Hayashi[1]

1 Department of Restorative Dentistry and Endodontology, Osaka University Graduate School of Dentistry, Osaka, Japan, 2 Unit of Dentistry, Osaka University Hospital, Suita, Osaka, Japan, 3 Department of Dentistry and Oral Maxillofacial Surgery, Hyogo College of Medicine, Nishinomiya, Hyogo, Japan, 4 Division of Materials and Manufacturing Science, Graduate School of Engineering, Osaka University, Suita, Osaka, Japan, 5 Department of Periodontology and Endodontology, Institute of Biomedical Sciences, Tokushima University Graduate School, Tokushima, Japan, 6 Division of Cariology, Operative Dentistry and Endodontics, Department of Oral Health Science, Niigata University Graduate School of Medical and Dental Sciences, Niigata, Japan

* maezono@dent.osaka-u.ac.jp

**Data Availability Statement:** All relevant data are within the paper and its Supporting Information files.

## Abstract

The purpose of this study was to develop a high-frequency wave therapy model in rats and to investigate the influence of high-frequency waves on root canal treatment, which may provide a novel strategy for treating apical periodontitis. Root canal treatments with and without high-frequency wave irradiation were performed on the mandibular first molars of 10-week-old male Wistar rats. The mesial roots were evaluated radiologically, bacteriologically, and immunohistochemically. At 3 weeks after root canal treatment, lesion volume had decreased significantly more in the irradiated group than in the non-irradiated group, indicating successful development of the high-frequency therapy model. The use of high-frequency waves provided no additional bactericidal effect after root canal treatment. However, high-frequency wave irradiation was found to promote healing of periapical lesions on the host side through increased expression of fibroblast growth factor 2 and transforming growth factor-β1 and could therefore be useful as an adjuvant nonsurgical treatment for apical periodontitis.

## Introduction

Apical periodontitis is a disease caused by bacterial infection in the root canal system [1]. Treatment methods for apical periodontitis include mechanical removal of the infection source in the root canal; chemical cleaning of the root canal with sodium hypochlorite (NaOCl) and ethylenediaminetetraacetic acid (EDTA); and dense, hermetically sealed obturation. Root canal treatment (RCT) can fail due to the anatomical complexity of the root canal system, which includes lateral branches and isthmuses [2], and extraradicular biofilms [3–6], which make it difficult to completely remove the source of bacterial infection [7]. The balance

**Funding:** This work was supported by JSPS KAKENHI Grant Numbers JP17K17129 (KK), JP19K19026 (NY), JP19K10107 (HM) and JP19K18994 (SM). The funders had no role in study design, data collection and analysis, decision to publish, or preparation of the manuscript.

**Competing interests:** The authors have declared that no competing interests exist.

between bacteria and the host immune system is a key element in biofilm-related infections [8]. Therefore, alternatives to removing the source of infection have been investigated, such as gene therapy [9–11]. Recent research has explored how strengthening the host immune system can lead to healing; this approach is the basis of the current study.

This study focused on high-frequency wave (HFW) apical therapy, an adjuvant nonsurgical treatment for apical periodontitis. HFW therapy is widely used as a diathermy therapy and has curative properties in human hard tissue, such as promoting bone union after fracture [12–14]. HFW irradiation generates heat as a result of vibration energy induced by free electrons colliding with molecules in a conductor as electric current flows. In dentistry, this principle is applied in electric cauterization [15, 16]. One indirect application of HFW is its use to promote bone healing; in Japan, HFW is applied in orthopedic surgery to treat slow-healing fractures and nonunion [17–21]. In the dental field, HFW therapy has also recently been used to promote healing after implant surgery [22].

HFW irradiation exhibits bactericidal effects on various free-floating bacteria, such as *Porphyromonas gingivalis*, *Staphylococcus intermedius*, *Enterococcus faecalis*, and *Streptococcus mutans* [23]. It can also induce gene expression of growth factors, such as transforming growth factor-β1 (TGF-β1) and fibroblast growth factor 2 (FGF2), in mouse osteoblasts [24].

In humans, HFW irradiation has been shown to significantly promote healing of periapical lesions [25]. However, the mechanism of action of HFW in the root canal—whether it targets the tissue around the apex, the bacteria, or both—remains unknown. The current study investigated the adjuvant effect of HFW irradiation in the nonsurgical treatment of apical periodontitis, which consisted of mechanical removal and chemical cleaning of the source of infection.

Recently, Yoneda et al. [26] developed a model for the treatment of infected root canals in rats. In this model, apical periodontitis was induced in rat mandibular first molars and RCT was then performed under rubber dam isolation. The volume of the periapical lesion and its change over time were observed with micro-computed tomography (micro-CT). RCT significantly reduced the bacterial count in the root canal, establishing a system leading to clinical healing. With this model it is possible to evaluate the changes and reactions occurring as a result of HFW irradiation after RCT.

In the present study, we developed a new high-frequency therapy model in rats by modifying the model of Yoneda et al. [26]. We used this new model to elucidate the mechanism of action of HFW therapy in apical periodontitis. The influence of HFW as a supplement to conventional RCT in treating periapical lesions was evaluated with three-dimensional micro-CT analysis, bacteriological evaluation, and immunohistochemical analysis.

## Materials and methods

### Ethics statement

This study was approved by the Animal Care and Use Committees of the Osaka University Graduate Schools of Dentistry and Engineering (Permit Nos. 26-016-0 and 26-1-0). All animal experiments were carried out in accordance with the Guidelines for Animal Experiments of Osaka University. Surgical procedures were performed under sodium pentobarbital anesthesia, and all efforts were made to minimize the animals' suffering.

### Animals

Thirty-four 10-week-old male Wistar rats (Clea Japan, Tokyo, Japan) were used in this study, as shown in Fig 1. The rats were divided into three groups: micro-CT (n = 12), bacterial quantification in the root canal (n = 10), and immunohistochemical observation (n = 12).

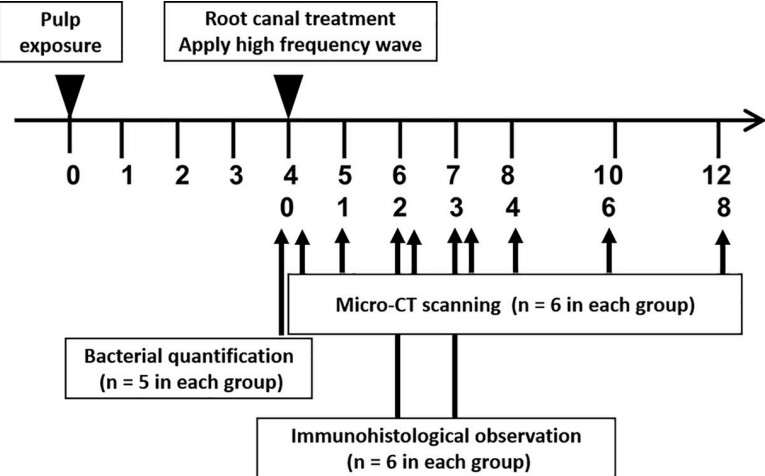

**Fig 1. Experimental schedule of high-frequency wave root canal therapy in rat model.**

The animals were maintained in the animal facility of the Osaka University Graduate School of Dentistry under a 12-h light/12-h dark cycle. Food and water were freely available. The sample size of each experimental group was set according to the method of Kadam et al. [27].

## High-frequency wave treatment device

HFW irradiation was performed with a prototype electrosurgical unit system (J. Morita Manufacturing, Kyoto, Japan) [23–25]. The system consists of a main body with a root canal length-measurement function, an active electrode, and a counter electrode. HFW irradiation is initiated by stepping on a foot pedal. A stainless steel #10 K-file (MANI, Tochigi, Japan) was used as the active electrode, and a stainless-steel hook was used as the counter electrode. Irradiation conditions were set according to the method of Tominaga et al. [25]: frequency 510 kHz, voltage 35 V, energization time 1 s, and irradiation interval 4 s.

## Root canal treatment in rats

Experimental periapical lesions were induced according to the methods of Kawahara et al. [28] and Kuremoto et al. [29]. Periapical lesions were induced by exposing the pulp of all right and left mandibular first molars. The rats were divided into an irradiated group and a non-irradiated group. In both groups, the root canals of the right mandibular first molars were prepared as in conventional RCT. RCT was performed according to the method of Yoneda et al. [26]. In short, the tooth was isolated with a custom-made rubber dam clamp (YDM, Tokyo, Japan) and rubber dam sheet (Heraeus Kulzer, Hanau, Germany). The pulp chamber was opened and necrotic coronal pulp was removed with a round bur and a micro-excavator (OK Micro-exca, Seto, Ibaraki, Japan). The working length was set to the level of 1.0 as indicated on an electronic apex locator (Root ZX, J. Morita Manufacturing) by using K-files (Dentsply Maillefer, Ballaigues, Switzerland); the root canal was then enlarged up to a #20 file. Root canals were irrigated with 0.5 ml each time, in total 2 ml of 2.5% NaOCl (Neo Dental Chemical Products, Tokyo, Japan) [30]. The canals were dried with sterilized paper points (VDW, Munich, Germany) and filled with #20 gutta-percha points (SybronEndo, Orange, CA, USA) and root canal sealer (RealSeal SE, SybronEndo) according to the single-cone obturation technique [31]. After processing with a bonding system (Clearfil Bond SE ONE, Kuraray Noritake

Dental, Tokyo, Japan), the pulp chamber was filled with flowable composite resin (MI FLOW, GC, Tokyo, Japan). In the irradiated group, HFW irradiation was applied once to the periapical lesion and twice to the inside of the root canal immediately before root canal obturation. The root canals of the non-irradiated group were obturated as in conventional RCT without HFW irradiation. The left mandibular first molars of the irradiated group were used as a control group without RCT.

## Three-dimensional measurement of periapical lesion volume

Twelve rats were divided equally into an irradiated group and a non-irradiated group (n = 6 in each group). At 0, 1, 2, 3, 4, 6, and 8 weeks after root canal filling, the periapical lesions of the rats were scanned with a micro-CT scanner (R_mCT2; Rigaku, Tokyo, Japan) under the following conditions: field of view: $\varphi 10 \times 10$ mm; voxel size: $20 \times 20 \times 20$ μm; tube voltage: 90 kV; and tube current: 160 μA. After scanning, 512 consecutive tomographic slice images, each with a thickness of 20 μm, were obtained. The images were then reconstructed with the Three-dimensional Reconstruction Imaging for Bone (TRI/3D-BON) system (Ratoc System Engineering, Tokyo, Japan). The volume of the periapical lesion at the mesial root was measured as previously described [26]. In short, a calibration curve for converting CT values into bone mineral density (BMD) values was prepared by using a phantom with a known BMD. Then, BMD images of each sample were obtained. The threshold for hard tissue extraction was set with discriminant analysis, and binarization was performed. The area of the apical lesion was measured on the binarized image. The space of the periodontal ligament around sound teeth in rats of the same age was determined. The apical lesion volume was obtained by subtracting the periodontal ligament space from all lesions in the apical area, and volume differences were compared among the groups.

## Quantification of bacteria in the root canal

Ten rats were divided into an irradiated group and a non-irradiated group (n = 5 in each group). All rats in both groups were euthanized immediately after RCT and the mandibular first molars were extracted. The teeth were cut at the mesial root furcation and bacteria were removed from the root surface by curetting with a sterilized spoon excavator (YDM). The mesial root was then frozen in liquid nitrogen and crushed with an SK mill (Tokken, Chiba, Japan). The samples were divided into two equal parts: one for the ATP assay to measure the number of viable bacteria in the root canal and the other for real-time polymerase chain reaction (PCR) to measure the total number of bacteria in the root canal. Following the method previously described by Yoneda et al. [26], DNA extraction was performed on a powdered sample with the InstaGene Matrix (Bio-Rad Laboratories, Hercules, CA, USA) according to the manufacturer's instructions. Assays were performed with a 20-μl solution containing 1 μl of DNA extract (Applied Biosystems Power SYBR Green PCR Master Mix; Life Technologies, Grand Island, NY, USA) and bacterial universal primers 357F and 907R [32] (0.5 μl each), which were prepared in parallel reaction mixtures for each target sequence. The thermal cycling conditions for the Applied Biosystems 7500 Fast Real-Time PCR system (Life Technologies) were 95°C for 10 min, 40 cycles at 95°C for 15 s, and 65°C for 1 min, with recording of the fluorescence signal at the end of each cycle. Melting-curve analysis consisted of a denaturation step at 95°C for 15 s and a temperature reduction to 60°C for 1 min followed by a temperature increase to 95°C at a rate of 1%, with continuous fluorescence reading. Data were acquired and analyzed with Applied Biosystems 7500 system SDS v2.0.2 software (Life Technologies). *Enterococcus faecalis* SS497 was used as a standard curve. The amount of ATP in the root canal, representing the number of live bacteria, was measured. The ATP eliminator kit

(AF-3X2; DKK-TOA, Tokyo, Japan) and ATP analyzer (AF-100; DKK-TOA) were used as previously described [33].

## Immunohistochemical observation

All 12 rats from both the irradiated group and the non-irradiated group (n = 6 in each group) were euthanized at 2 or 3 weeks after root canal filling. Pentobarbital sodium was intraperitoneally administered, and perfusion fixation was performed with periodate lysine paraformaldehyde (PLP) fixative (Wako Pure Chemical Industries, Osaka, Japan). Mandibular samples containing the first molars were dissected, fixed in PLP for 12 h at 4°C, and decalcified in 10% EDTA containing 15% glycerol at 4°C. Serial sections of 5-μm thickness were stained with an enzyme antibody method using specific antibodies against interleukin-1β (IL-1β) (ab 9787; Abcam, Cambridge, England), TGF-β1 (MAB 240; R & D systems, Minneapolis, MN, USA), and FGF2 (ab 16828; Abcam) and then observed under a light microscope (Optiphot-2; Nikon, Tokyo, Japan).

## Statistical analysis

Tukey's test was used to detect statistically significant differences in the volume change of the periapical lesions. The Steel–Dwass test was used for statistical analysis of the quantification of root canal bacteria. Each risk factor was evaluated as 5%. IBM SPSS Statistics (version 22.0, IBM SPSS, Chicago, IL, USA) was used for statistical analyses.

## Results

### Three-dimensional measurement of periapical lesion volume

Representative micro-CT images of the buccolingual cross-section of the mesial root of the mandibular first molar at each timepoint are shown in Fig 2A. Compared with the control group, the volume of periapical lesions in the mesial root in the irradiated group was significantly smaller at 2 weeks after RCT (p = 0.009) and at 6 weeks after RCT (p = 0.013; Fig 2B). Furthermore, the lesion volume had decreased significantly more at 3 weeks after RCT in the irradiated group than in the non-irradiated group (p = 0.019; Fig 2B).

### Quantification of bacteria in root canal

The number of live bacteria and the total number of bacteria in the root canal were significantly lower in both the irradiated and non-irradiated groups, compared with numbers in the control group (p = 0.043 and 0.035, respectively). No significant difference in bacterial numbers was observed between the irradiated group and the non-irradiated group (**Fig 3**).

Immediately after RCT, (a) the number of live bacteria in the root canal after high frequency wave irradiation was quantified with an ATP assay and (b) the number of total bacteria was quantified with real-time PCR. The number of bacteria in the root canal of the left first molar of rats in the irradiated group was simultaneously determined as the control group. The graph shows the average value and standard deviation. Significant differences can be seen between different characteristics in the same graph (n = 5 each, Steel–Dwass test, p < 0.05).

### Immunohistochemical analysis of periapical lesions

At 2 weeks after RCT, IL-1β-positive cells were widely expressed in the periapical lesions in both the irradiated group and the non-irradiated group (Fig 4A, 4C, 4E and 4G). However, at 3 weeks after RCT, the expression was limited in both groups (Fig 4B, 4D, 4F and 4H). Additionally, the expression of IL-1β decreased with HFW irradiation (Fig 4B, 4D, 4F and 4H). At 3

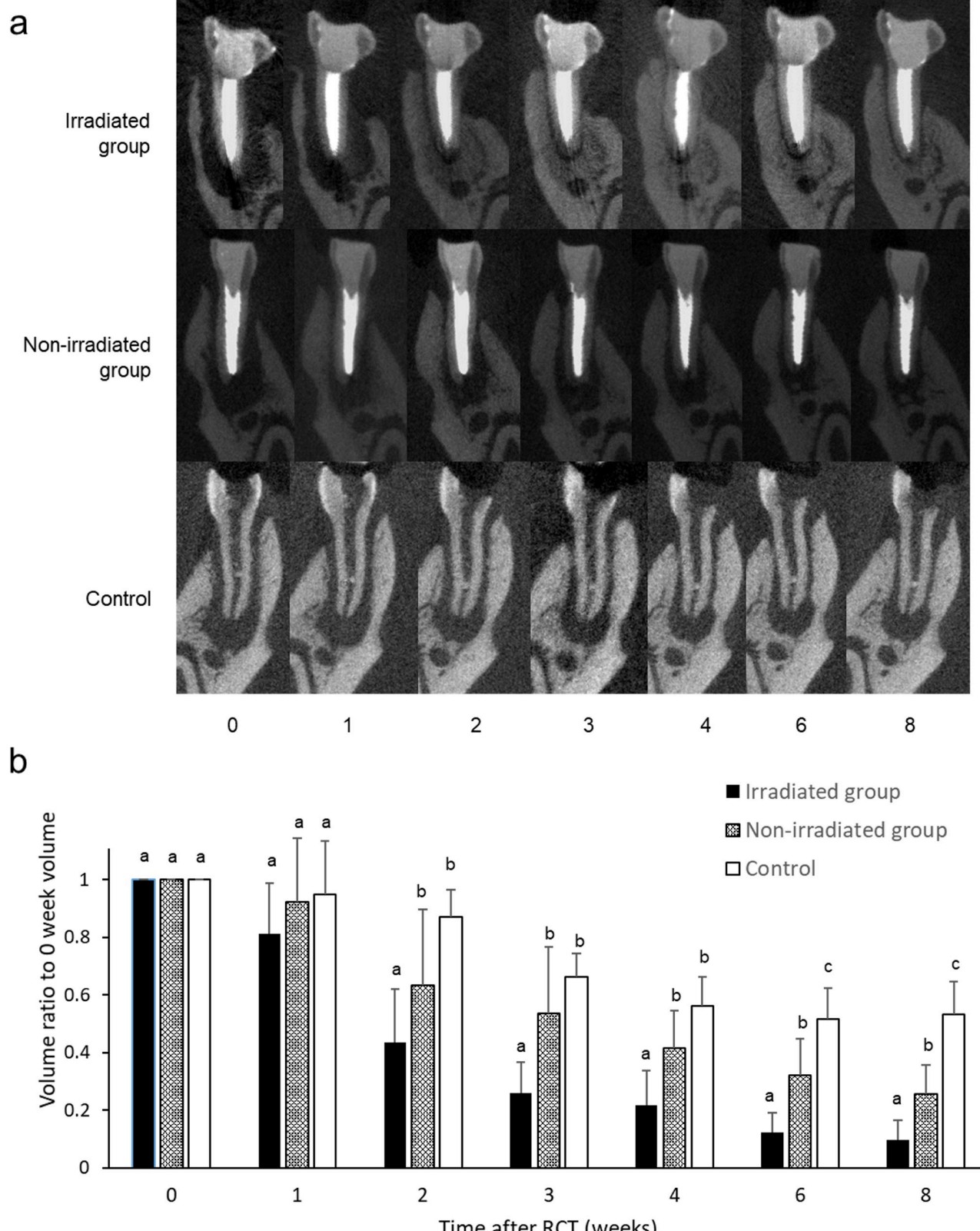

**Fig 2. Three-dimensional measurement of periapical lesion volume.** a: Representative micro-CT images of mesial root of mandibular first molar at each timepoint. b: The volume of the periapical lesion of the mesial root was measured at each timepoint (n = 6 each). The graph shows the mean and

standard deviation of the ratio of the volume at each timepoint to the volume at week 0 after root canal treatment (RCT). Significant differences are represented by different letters in the same week group (Tukey's test, p < 0.05).

weeks after RCT, there was more expression of FGF2-positive cells within the periapical lesions in the irradiated group than in the non-irradiated group (Fig 4J, 4I, 4N and 4P). The irradiated group showed more expression of TGF-β1-positive cells at the boundary between the periapical lesion and the alveolar bone at 3 weeks after RCT, compared with the non-irradiated group (Fig 4R, 4T, 4V and 4X).

## Discussion

This work assessed the influence of HFW irradiation in vivo in a rat model and investigated the target of HFW.

Studies of HFW therapy have focused on the disinfection properties of HFW [23] and its promotion of tissue regeneration [17–22]. In human clinical studies, healing of periapical lesions is seen by 1 month after RCT [25]. In our study, the lesion volume in the irradiated group had decreased significantly at 3 weeks after RCT with HFW irradiation (Fig 2). Human studies have also shown that HFW consistently promotes healing within a certain time period following RCT [25]. Other research has suggested that bone regeneration in skull defects may be promoted by irradiation with 510 kHz HFW, and that excessive irradiation conversely inhibits bone formation [34]. In the present study, we used a frequency of 510 kHz, which is the same frequency as that used for electric cauterization. In previous reports, no differences were found in the bactericidal effect of HFW at frequencies above 500 kHz [23].

It has been reported that HFW exerts a bactericidal effect against free-floating bacteria [23]. The mechanism by which HFW stimulation achieves sterilization may be related to thermal

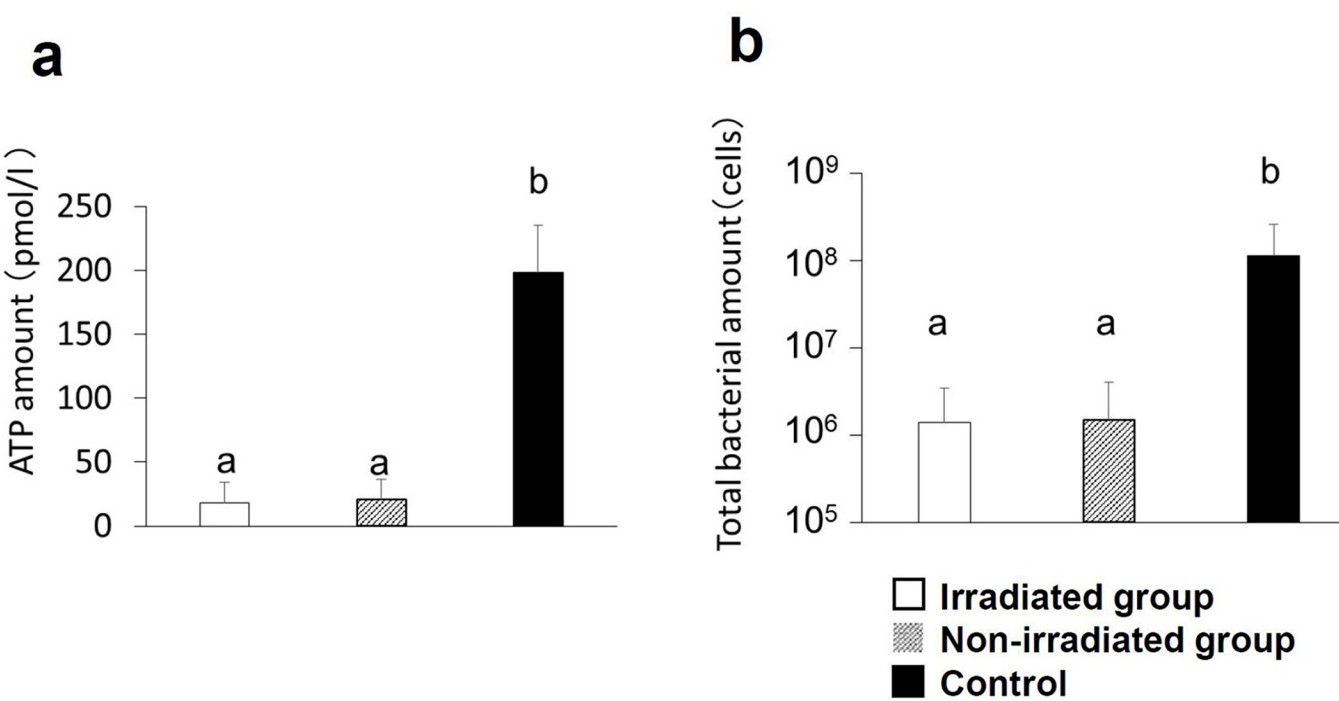

**Fig 3. Quantification of bacteria in root canal.** a: ATP level (number of live bacteria) in root canal immediately after root canal treatment (RCT). b: Total number of bacteria in root canal after RCT.

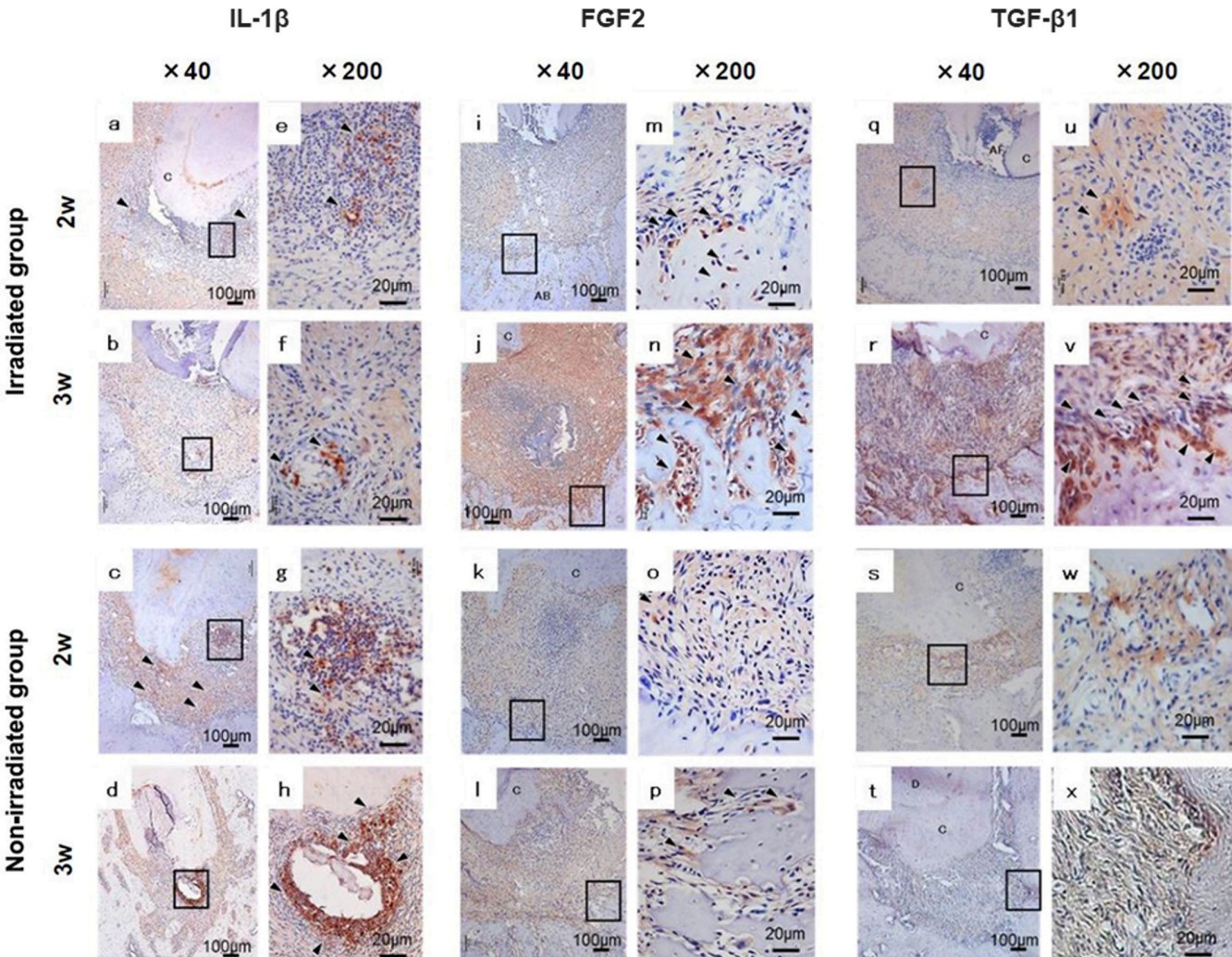

**Fig 4. Immunohistochemical analysis of mesial root lesions.** Immunohistochemical analysis was performed for IL-1β (**a–h**), FGF2 (**i–p**), and TGF-β1 (**q–x**) in the periapical area at 2 weeks (**a, c, e, g, i, k, m, o, q, s, u, w**) and 3 weeks (**b, d, f, h, j, l, n, p, r, t, v, x**) after root canal treatment. Panels e–h, m–p, and u–x show high-magnification views of the framed areas in panels a–d, i–l, and q–t, respectively. AB: alveolar bone, C: cementum, D: dentin, P: pulp, arrowheads: positive cells.

energy and electrical energy. Because joule heat is generated by energizing, thermal energy is generated during HFW irradiation. The temperature rise resulting from a single HFW irradiation of 125 µl of phosphate-buffered saline is approximately 4.1˚C; the temperature of the liquid drops to the original temperature after an interval of 4 s [35]. In the present study, no adverse events were observed throughout the observation period. Similarly, no adverse events were reported in the human clinical study by Tominaga et al. [25]. Therefore, HFW irradiation and the associated temperature increase were determined not to be harmful under the conditions tested; this finding suggests the possibility of clinical application of the procedure as adjunctive therapy for apical periodontitis.

No significant change was observed in the number of bacteria after HFW irradiation during RCT in the rat model (Fig 3). The reason for this lack of change may be that it is difficult to irradiate the biofilm directly inside the root canal and the apex. Furthermore, residual bacteria decreased significantly after RCT alone, compared with the level before treatment. Yoneda

et al. reported that RCT removed 75% of bacteria in the root canal and that 75% of cases were cured with conventional RCT; however, the remaining 25% of bacteria were difficult to remove [26]. Our findings suggest that HFW irradiation had no effect on the number of remaining bacteria in this experimental setting. However, because of differences in physical size, a lower voltage (35 V) was used in rats compared with that used in humans; therefore, the treatment may have been compromised. HFW irradiation at a higher voltage may have bactericidal effects.

Yumoto et al. [24] reported that HFW irradiation induced gene expression of growth factors, such as TGF-β1, VEGF, and FGF2, in mouse osteoblasts in vitro. In the present study, the expression of TGF-β1 was upregulated at the border between the alveolar bone and the lesion in the HFW-irradiated group (Fig 4R and 4V); FGF2 expression was also upregulated within the lesion (Fig 4J and 4N). These data suggest that HFW promoted the healing of apex lesions by promoting the expression of TGF-β1 and FGF2. In addition, IL-1β was downregulated in the HFW-irradiated group, suggesting that HFW irradiation suppressed inflammation in the periapical lesions. Further evaluation of the expression of osteoblastic markers and proliferating cells in the lesions is needed.

For the treatment of biofilm infections, it is critical to remove the infectious source mechanically or chemically; however, it is impossible to completely remove the source from the interior and exterior of the root canal, as previously indicated. In this study, HFW irradiation promoted the healing of apical lesions via interaction with the host side. Therefore, HFW irradiation offers the possibility of promoting healing through enhancement of the host immune system in apical lesions after some of the infectious source is successfully removed through RCT.

Much research has been conducted on biofilm infections; however, these infections arise from a complex variety of factors, many of which have not yet been fully elucidated. Our study focused on the essential conditions in situ/in vivo, and our findings indicate that HFW may be effective in stimulating immune-response cells in a rat model for treatment of infected root canals. Of particular interest is the effect of HFW on host cells. Therefore, we suggest that RCT with HFW may increase healing speed and thus merits development as a new adjuvant treatment for apical periodontitis.

## Acknowledgments

We thank Kazunari Matoba from J. Morita Manufacturing for providing the high-frequency wave equipment; Naoto Ohkura, DDS, PhD, from Niigata University for technical assistance with the immunohistochemical analysis.

## Author Contributions

**Conceptualization:** Katsutaka Kuremoto, Hiromichi Yumoto, Shigeyuki Ebisu, Yuichiro Noiri.

**Data curation:** Saori Matsui, Naomichi Yoneda.

**Formal analysis:** Yuichiro Noiri.

**Funding acquisition:** Saori Matsui, Naomichi Yoneda, Hazuki Maezono, Katsutaka Kuremoto.

**Investigation:** Saori Matsui, Naomichi Yoneda.

**Methodology:** Saori Matsui, Hazuki Maezono, Takuya Ishimoto, Takayoshi Nakano, Hiromichi Yumoto, Yuichiro Noiri.

**Project administration:** Hazuki Maezono, Yuichiro Noiri.

**Resources:** Takuya Ishimoto, Takayoshi Nakano.

**Supervision:** Shigeyuki Ebisu, Yuichiro Noiri, Mikako Hayashi.

**Validation:** Hiromichi Yumoto.

**Visualization:** Saori Matsui.

**Writing – original draft:** Saori Matsui, Naomichi Yoneda, Hazuki Maezono.

**Writing – review & editing:** Saori Matsui, Naomichi Yoneda, Hazuki Maezono, Yuichiro Noiri, Mikako Hayashi.

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
