## [Decision Letter · Decision Letter 0]

27 Apr 2020

PONE-D-20-08421

Assessment of the functional efficacy of root canal treatment with high frequency waves in rats

PLOS ONE

Dear Dr Maezono,

Thank you for submitting your manuscript to PLOS ONE. After careful consideration, we feel that it has merit but does not fully meet PLOS ONE’s publication criteria as it currently stands. Therefore, we invite you to submit a revised version of the manuscript that addresses the points raised during the review process.

The manuscript while is interesting and may merit publication in PLoS ONE, it needs a lot of work. There are several clarifications (see comments from reviewers). More importantly, we need to make a full data available (see reviewer #1). 

We would appreciate receiving your revised manuscript by Jun 11 2020 11:59PM. To enhance the reproducibility of your results, we recommend that if applicable you deposit your laboratory protocols in protocols.io, where a protocol can be assigned its own identifier (DOI) such that it can be cited independently in the future. For instructions see: http://journals.plos.org/plosone/s/submission-guidelines#loc-laboratory-protocols

We look forward to receiving your revised manuscript.

Kind regards,

Sompop Bencharit, DDS, MS, PhD, FACP

Academic Editor

PLOS ONE

Additional Editor Comments:

Two reviewers recommended reject; however, the other two recommended Revision. The main concerns were the data analysis and availability of the entire data as well as several clarifications throughout the manuscript.

Reviewers' comments:

Reviewer's Responses to Questions

**Comments to the Author**

1. Is the manuscript technically sound, and do the data support the conclusions?

Reviewer #1: No

Reviewer #2: Partly

Reviewer #3: Yes

Reviewer #4: Yes

Reviewer #5: Partly

2. Has the statistical analysis been performed appropriately and rigorously? 

Reviewer #1: No

Reviewer #2: I Don't Know

Reviewer #3: Yes

Reviewer #4: Yes

Reviewer #5: I Don't Know

3. Have the authors made all data underlying the findings in their manuscript fully available?

Reviewer #1: Yes

Reviewer #2: Yes

Reviewer #3: Yes

Reviewer #4: No

Reviewer #5: Yes

4. Is the manuscript presented in an intelligible fashion and written in standard English?

Reviewer #1: No

Reviewer #2: No

Reviewer #3: No

Reviewer #4: Yes

Reviewer #5: Yes

5. Review Comments to the Author

Reviewer #1: The purpose of this study was to investigate the potential effects of high frequency wave (HFW) irradiation in healing outcomes of experimental apical periodontitis in vivo and in vitro conditions. Several methodological concerns and lack of clarity in details jeopardize proper interpretation of study findings. The results need careful interpretation in light of the lack of translatability of methods to be used in clinical practice, and the lack of a true biofilm in the experiments. It is not clear why the authors went into great lengths with the in vivo study to then turn into in vitro studies to assess expression of growth factors. It would make more sense to use periapical lesions/normal tissues from the animals for these assays.

Extensive grammatical and technical revisions are necessary.

line 48 – What are the authors trying to say with: “It has been reported that the average success rate for initial root canal treatment (RCT) was lower than that of pulpectomy [2–5].” Pulpectomy is part of RCT and these procedures are not alternatives of each other. This does not make sense.

Line 48- Revise ‘tight’ obturation with ‘dense, hermetically-sealed’ obturation

Line 58 – a reference is needed at the end of the sentence

Line 59 – HFW is NOT an accepted method for NSRCT, therefore the authors should limit their purpose to analyse HFW as an ADJUVANT in NSRCT procedures.

Line 113 – add some brief details of the lesion induction protocol for clarity. Were lesions induced on the right and left side of all animals in both irradiated and non irradiated groups? Clarify

Lines 118-120 – Please revise this sentence. “enlarging’ a canal means the apical diameter established during instrumentation. I assume “enlarged to the level of 1.0” refers to working length distance to the apical foramen, as determined by an electronic apex locator (which the authors erroneously call ‘root canal meter’. The size of apical enlargement, which should be standardized between groups and of utmost importance to this study however is not mentioned in the text.

Justify the use of 0.5ml of 2.5% NaOCl

Justify the use of single cone obturation technique and provide details of the gutta percha cones used

Line 126, add number of rats divided in irradiated and non irradiated groups each

Line 128 – HFW was irradiated – was this through the foramen? How far out into the periapical lesion and how was this measured? Clarify

Line 203 – Biofilm formation was described to occur using 7-day polymicrobial cultures. However true biofilms have been shown to require a minimum of 21 days of culture. Hence the authors cannot state that a biofilm was used, only a polymicrobial mixture.

The results presented do not provide clear direction as if the images in Fig. 2 and 4 come from one animal or if representative sections of various animals are presented.

Line 351 – this does not reflect the study purpose stated in the introduction

The conclusions are not supported by the data presented – e.g. “it was proved that HFW irradiation promoted periapical lesion healing”. The authors fail to recognize that the success of endodontic treatment alone is >90%, therefore the HFW if any, might have helped the process by stimulating immune-response cells.

Lastly, HFW cannot be suggested as a new treatment for apical periodontitis as standalone. Root canal treatment is still necessary, as shown by their own results in which HFW had little to no activity on bacteria. Therefore the conclusions need to be toned down.

Figure 1 – add the number of animals used in each experiment

Figure 2 – include representative microCT images for each test and control site in irradiated and non irradiated groups that correlate with the results graph presented. Ideally the microCT images in which the obturation of the root canal can be visible as well as the periapical lesion size must be included.

Figure 4 – include lower magnification images showing the level of root canal obturation in irradiated and non irradiated groups.

Reviewer #2: I would like to thank the authors for submitting their article to PLOS ONE. It was my pleasure to review the manuscript entitled “Assessment of the functional efficacy of root canal treatment with high frequency waves in rats”. Undoubtedly, a lot of work was done to carry out this study and I commend the authors on their efforts in preparing the manuscript. Several analyses were performed in this study, mixing in vivo and in vitro assays, which makes it too long, difficult to read and sometimes even confusing. I suggest the authors to divide this paper in two: in vivo and in vitro assessment. Although the manuscript is very interesting, there are significant shortcomings that deter my enthusiasm.

- Overall, the entire manuscript needs to be revised for proper English/grammar/syntax/spelling

Introduction

- "RCT often 50 fails because the anatomical complexity of the root canal system, which includes lateral branches and isthmuses". This is information is not correct.. the success rate of primary root canal treatment is high, around 90-95%

- "This study focused on high frequency wave (HFW) apical therapy, which is a nonsurgical treatment for apical periodontitis" This kind of sentence should not be in this section

Methods

- This section is confusing and also the experiments need to be better described instead of using the expression "according to the method"

- Did the authors measured the periapical lesions before root canal treatment?

- Page 8, line 118. Eletronic apex locator instead of electrical root canal meter

- Which file was used to enlarge the root canals?

- Page 9, line 135. "the periapical lesions were scanned with a micro-135 CT scanner" the rats were scanned, not the periapical lesions"

- What were the parameters used during the scanning procedure?

- What about the radiation in which these rats where submitted during all these scans?

Results

- Is there a control group in the micro-ct analysis? The authors did not mention it on the MM section. What is the N of this group?

- This section also needs to be rewritten

Discussion

- This section is poorly written

- The authors did not discuss the results of immunohistochemical analysis of the periapical lesion and the quantification of growth factors expression with rat osteoblasts and fibroblasts

Figures

- A micro-CT image showing the reduction of the periapical lesion is relevant to present herein

Reviewer #3: The use of high frequency waves in endodontic treatment is a hot topic nowadays and this study was well designed. This manuscript focused on the assessment of the functional efficacy of RCT with HFW in vivo with rat model and also in vitro. I appreciate the hard work of the authors to provide important information to the endodontic field regarding HFW.

I have some recommendations and concerns:

1. In introduction line 48, why did authors compared the success rate for initial root canal treatment with pulpectomy? What was the purpose of the comparison of these two procedures?

2. Please check grammar and abbreviations, eg. line 76 micro-computed tomography should be (Micro-CT)

3. The description of materials and methods needs to be reorganized. The paragraph describing root canal treatment in rats (line 107-131) was unclear and was repeated in the paragraph of quantification of bacteria in the root canal (line 148-173)

4. The paragraph of statistical analysis (line 246-252) needs to be revised. In scientific writing, authors should avoid using "we used...".

5. In the discussion, authors discussed the mechanism of HFW's bactericidal effect and its thermal energy generation. Could authors discuss the potential damage of periodontal ligament by using HFW given to the rise of temperature?

Again, I appreciate the authors' contribution and looking forward to your responses.

Reviewer #4: The aim of this paper is interesting and the study may provide a new non-surgical treatment for apical periodontitis, however some flaws need to be addressed.

Abstract:

1. Line 38-40, the authors concluded that “high frequency wave irradiation promoted healing of the periapical lesion through increased expression of VEGF, FGF2 and TGF-β1”,however, in the part of ”Immunohistochemical observation” (methods), the samples were stained with specific antibodies against IL-1β, TGF-β1 and FGF2. Please explainhow these results lead to the conclusion described above.

Introduction:

2. Line 48-49, “It has been reported that the average success rate for initial root canal treatment (RCT) was lower than that of pulpectomy”. This sentence is quite confusing. Please explain this in detail.

Methods:

3. In the parts of “Root canal treatment in rats”, line 128-129, “HFW irradiation was applied three times each to the periapical lesion and the inside of the root canal immediately before the root canal obturation”, however, in the part of “Quantification of bacteria in the root canal”, line 150-151, “HFW irradiation was applied three timesto the periapical lesion and the root canal after the root canal preparation”. Are these two time point the same?

4. Line 153-154, “Immediately following the RCT, 10 rats were sacrificed and the mandibular first molars were extracted”, the rats were sacrificed and bacteria were collected immediately following the RCT, however, in the corresponding part of result, line 267, “At 4 weeks after RCT”. Authors described the ATP and PCR results for “Quantification of bacteria in the root canal” was from samples from 4 weeks after RCT. Please confirm the descriptions and make the unification.

Results:

5. Line 255-258, “Compared with the controlgroup, the volume of the periapical lesions in the mesial root was significantly smaller in the irradiated group at 2 weeks after the RCT (p = 0.009), and in the non-irradiated group at 6 weeks after the RCT(p = 0.013, Fig. 2)”, authors described that the volume of periapical lesions was significantly smaller in non-irradiated group at 6 weeks after RCT, but form Fig. 2, we could find that at 2 weeks after RCT, the volume of periapical lesions was also significantly smaller in non-irradiated group when compared with the control group (“a” and “b” meant significant difference). Please confirm the result.

6. Line 333-335, “In rat osteoblasts, the expression of FGF2 was significantly higher in HFW irradiated osteoblasts than in non-irradiated osteoblasts at 3 days (irradiation applied 5 and 10 times, p = 0.010) and 5 days (irradiation applied 10 times, p = 0.013)”, the authors obtained the result that the expression of FGF2 was significantly higher in HFW irradiated osteoblasts than in non-irradiated osteoblasts at 5 days with 10 times irradiation, however, from Fig. 7(b), we could observed that both for irradiation with 5 times and 10 times at 5 days, expressions of FGF2 showed no significant difference (the same letter “a” meant no significant difference). Please confirm the result.

7. Line 344-345, “By comparing the expression level of (a) FGF2, (b) VEGF, and (c) TGF-β1 in rat fibroblasts, and (d) FGF2, (e) VEGF, and (f) TGF-β1 in rat osteoblasts”, the authors described that (a), (b) and (c) meant the expression level in rat fibroblasts, and (d), (e) and (f) meant the expression level in rat osteoblasts. However, in Fig. 7, we found t (a), (b) and (c) meant the expression level in rat osteoblasts, and (d), (e) and (f) meant the expression level in rat fibroblasts, which was quite contrary to the description. Please confirm it.

Discussion:

8. The irradiation times in all parts of in vivo were consistent (three times each to the periapical lesion and the inside of the root canal), however, in the in vitro assays, the irradiation times were various (15 times for biofilm discs, 5 and 10 times for cells). The authors should discuss why they chose these different irradiation times and whether the different irradiation times affected the therapeutic effect.

Reviewer #5: The central idea of the manuscript was to access antibacterial/antibiofilm efficacy and growth factors expression to high frequency waves applied during endodontic treatment in teeth with periapical lesions. The manuscript proposal is interesting and focuses on a new possibility for root canal treatment. However, some major changes should be considered.

Abstract

1) Objective is not clear – to “investigate the influence of root canal treatment using high-frequency waves” on…? Antibiofilm effect was not mentioned. Growth factors expression was note mentioned.

2) Methodology should be better explained. It is too superficial. It is not possible to understand how the in vitro e in vivo experiments were performed. It is too confusing.

Introduction

1) Introduction should be more concise and objective.

2) Authors reported: “Recently, it was reported that the balance between bacteria and the host immune system is important for biofilm related infections [13].” Recently, from 2002? It is not recent information.

3) The reference is missing: “The initial choice for the treatment of apical periodontitis is to remove the source of infection; however, it is speculated that strengthening the host immune system can also lead to healing.”

4) Authors reported: “This study focused on high frequency wave (HFW) apical therapy, which is a nonsurgical treatment for apical periodontitis.” Is HFW a nonsurgical treatment for apical periodontitis? Reference?

5) Authors reported: “In the dental field, this principle is applied in electric cauterization.” Reference?

6) TGF-β1, VEGF and FGF2. Abbreviation should be described first.

Materials and Methods

1) Authors reported: “Root canal treatment in rats: Thirty-four 10-week-old male Wistar rats were used for this experiment, and the experimental design is shown in Figure 1.” Information about rats had already been mentioned - unnecessary.

2) Figure 1 should be mentioned before the topic “root canal treatment in rats”.

3) Why only the left mandibular first molars of the irradiated group were used as control group without RCT?

4) Authors reported: “In both groups, the root canals of the right mandibular first molars were prepared as in conventional RCT.” Which groups?

5) What was the tip of the endodontic file used?

6) Both topics “Root canal treatment in rats” and “Quantification of bacteria in the root canal” present information about root canal treatment and HFW.

7) About the topic “Biofilm formation” – “the HA discs…had been treated with saliva for more than 8 h.” Sterile saliva? Normal saliva? More than 8h means 9h, 10h, 24h?? Number of samples?

8) “S. mutans and E. faecalis were cultured for 2 days, and P. intermedia, P. gingivalis and F. nucleatum were cultured for 7 days.” What does it mean?

9) Authors used different times of HFW irradiation for different experiments (root canal and periapical lesions, biofilm on HA discs, rat cells). Why?

Results/figure

1) “The effect of HFW radiation on bacterial biofilm in vitro: Observation of the biofilm on a hydroxyapatite (HA) disc with CLSM revealed that P. gingivalis and P. intermedia showed an antibiofilm effect after direct HFW irradiation (Fig. 5a)”. It should be better described. Bacterial species showed antibiofilm effect or HFW irradiation promoted an antibiofilm effect on P. gingivalis and P. intermedia?

2) Figure

Discussion

1) “This work assessed the influence of HFW irradiation by studying the general frequency, voltage, energization time, and interval of HFW.” It was not the objective of the present study. Authors did not test different frequencies, voltage, energization time and interval…at least it was not mentioned at the methodology. According to HFW treatment device: “Irradiation conditions were set according to the method of Tominaga et al: frequency 510 kHz, voltage 35 V, energization time 1 s, irradiation interval 4 s.” Authors only varied the number of application.

2) Authors reported: “In previous human studies, as in this rat study, the promotion of healing by HFW after a certain time period following RCT has been consistent [20].” However, reference 20 is a review article related to bone fracture and the conclusion refers to the uncertainty surrounding the use of electrical stimulation and fracture healing.

3) Authors did not mention anything about possible side effects or consequences of the HFW.

References

1) Most references are out of date.

6. PLOS authors have the option to publish the peer review history of their article (what does this mean?). If published, this will include your full peer review and any attached files.

Reviewer #1: No

Reviewer #2: No

Reviewer #3: No

Reviewer #4: No

Reviewer #5: No

---

## [Author Response · Author response to Decision Letter 0]

11 Jun 2020

Thank you very much for your e-mail of April 28th, 2020 concerning our manuscript Submission ID [PONE-D-20-08421] - [EMID:97f63f5e89367a4c] entitled " Assessment of the functional efficacy of root canal treatment with high frequency waves in rats ".

According to the suggestions, we modified the text and figures. We also appreciate the comments made by the editors and the reviewers. As reviewers pointed out, our in vitro experimental results caused some confusion so we clarified the points by focusing on in vivo experiments and modifying the body substantially. The point-by-point responses to each question or suggestion raised by the reviewers are attached as another file. In revising the manuscript, we paid much attention to the points suggested. We hope that these responses will prove satisfactory.

---

## [Decision Letter · Decision Letter 1]

15 Jul 2020

PONE-D-20-08421R1

Assessment of the functional efficacy of root canal treatment with high frequency waves in rats

PLOS ONE

Dear Dr. Maezono,

Thank you for submitting your manuscript to PLOS ONE. After careful consideration, we feel that it has merit but does not fully meet PLOS ONE’s publication criteria as it currently stands. Therefore, we invite you to submit a revised version of the manuscript that addresses the points raised during the review process.

While the revised manuscript is significantly improved, there are some clarifications needed throughout the manuscript. There are some wordings and clarification in all sections of the manuscript. Please review the comments from the reviewers and address them accordingly.

We look forward to receiving your revised manuscript.

Kind regards,

Sompop Bencharit, DDS, MS, PhD, FACP

Academic Editor

PLOS ONE

Additional Editor Comments (if provided):

While the revised manuscript is significantly improved, there are some clarifications needed throughout the manuscript. There are some wordings and clarification in all sections of the manuscript. Please review the comments from the reviewers and address them accordingly.

Reviewers' comments:

Reviewer's Responses to Questions

**Comments to the Author**

1. If the authors have adequately addressed your comments raised in a previous round of review and you feel that this manuscript is now acceptable for publication, you may indicate that here to bypass the “Comments to the Author” section, enter your conflict of interest statement in the “Confidential to Editor” section, and submit your "Accept" recommendation.

Reviewer #2: All comments have been addressed

Reviewer #3: All comments have been addressed

Reviewer #4: All comments have been addressed

2. Is the manuscript technically sound, and do the data support the conclusions?

Reviewer #2: Partly

Reviewer #3: Yes

Reviewer #4: Yes

3. Has the statistical analysis been performed appropriately and rigorously? 

Reviewer #2: Yes

Reviewer #3: Yes

Reviewer #4: N/A

4. Have the authors made all data underlying the findings in their manuscript fully available?

Reviewer #2: Yes

Reviewer #3: Yes

Reviewer #4: Yes

5. Is the manuscript presented in an intelligible fashion and written in standard English?

Reviewer #2: No

Reviewer #3: Yes

Reviewer #4: Yes

6. Review Comments to the Author

Reviewer #2: It was my pleasure to revise again this paper. The authors tried to answer all the questions raised by the reviewers, which improved the quality of this paper. Therefore, I commend the authors on their efforts. However, I still have several concerns/suggestions that need to be adressed

1. Overall, English still needs to be revised

2. Abstract: this section is confusing. I suggest the authors to start this section with the aim of the manuscript. Then, explain, in a concise way, the methods used.

3. Introdution: another section that needs to be revised in order to make it

- Page 9, line 52. Root canal system instead of root canal

- Page 9, line 54. Remove hand and rotary files as the root canal can be prepared using other kinematics as reciprocation, oscillatory

- Page 9, line 56. Remove the word often

- Page 10, line 60. "Removing the source of infection is considered the choice treatment" The authors mentioned it a few line above

4. Methods:

- Please provide the total amount of irrigation used per tooth (the authors mentioned it in a reply to a reviewer

- Why did the authors use RealSeal? This is not fabricated anymore

- How was the HFW irradiation applied to the periapical lesion?

- How was the procedure to assess the same cross-section at the same point in all time-points? Did the authors register the image-stacks?

5. Discussion:

- Page 20, line 246. This is not correct, as this was not the aim of this paper. Please, rewrite it. Besides, do not use the word biofilm

- Page 22, line 271. "RCT successfully removes most bacteria leaving only that which has invaded deep inside the dentinal tubules". This is incorrect, as several studies already demonstrated that 30-60% of the root canal walls remain untouched after root canal preparation

Reviewer #3: Thank you for revising your manuscript and your hardworking is appreciated by the reviewer.

Although the authors addressed most of the previous comments from the reviewer, it is still unclear in the M&M session.

For example, in each paragraph, authors used the same expression of " Twelve rats were divided equally into an irradiated group (n = 6) and a non-irradiated group (n = 6)" twice, and "The rats were divided into an irradiated group (n = 5) and a non-irradiated group (n = 5)." once.

I understand that there were 34 rats in total included in this study, and they were divided into three groups: MicroCT, Quantification of bacteria in the root canal, and Immunohistochemical observation. For each method, there were 12, 10, and 12 subjects included, respectively.

However, the M&M was still not clear and the repetitive description of "Twelve rats were divided equally into an irradiated group (n = 6) and a non-irradiated group (n = 6)" will cause confusion and may require readers to go back and forth to check the subject allocation.

Please revise this session accordingly.

Also, the sentence in Line 55 "treatment (RCT) often fails because the anatomical complexity of the root canal system, which includes lateral branches and isthmuses [2] and extraradicular biofilms [3-6], makes it difficult to completely remove the source of the bacterial infection [7]." is still inappropriate. The meaning of this sentence is inexplicit because it can be interpreted as "RCT often fails". In fact, initial NSRCT has a success rate around 95%.

Reviewer #4: (No Response)

7. PLOS authors have the option to publish the peer review history of their article (what does this mean?). If published, this will include your full peer review and any attached files.

Reviewer #2: No

Reviewer #3: No

Reviewer #4: No

---

## [Author Response · Author response to Decision Letter 1]

27 Aug 2020

Reviewer #2: It was my pleasure to revise again this paper. The authors tried to answer all the questions raised by the reviewers, which improved the quality of this paper. Therefore, I commend the authors on their efforts. However, I still have several concerns/suggestions that need to be adressed

1. Overall, English still needs to be revised

Proofreading by an English native speaker was performed and attached please find the certificate of proofreading.

2. Abstract: this section is confusing. I suggest the authors to start this section with the aim of the manuscript. Then, explain, in a concise way, the methods used.

Following your suggestion, we changed abstract to start with the aim, and then explained the methods used more concisely.

3. Introdution: another section that needs to be revised in order to make it

- Page 9, line 52. Root canal system instead of root canal

We changed the sentence following your advice.

- Page 9, line 54. Remove hand and rotary files as the root canal can be prepared using other kinematics as reciprocation, oscillatory

We removed the phrase “hand and rotary files” following your suggestion.

- Page 9, line 56. Remove the word often

We removed the word “often” following your advice.

- Page 10, line 60. "Removing the source of infection is considered the choice treatment" The authors mentioned it a few line above.

We deleted that part to avoid duplication.

4. Methods:

- Please provide the total amount of irrigation used per tooth (the authors mentioned it in a reply to a reviewer

We revised this part as below.

“Root canals were irrigated with 0.5 ml each time, in total 2ml of 2.5% NaOCl.”

- Why did the authors use RealSeal? This is not fabricated anymore

Real Seal SE (Pentron) was used in this experiment because it was used in the previous study by Yoneda et al. Real Seal SE sealer is currently available for purchase in Japan.

- How was the HFW irradiation applied to the periapical lesion?

We applied once to the periapical lesion and twice to the inside of the root canal as described in materials and methods section as below.

“HFW irradiation was applied once to the periapical lesion and twice to the inside of the root canal immediately before root canal obturation.”

- How was the procedure to assess the same cross-section at the same point in all time-points? Did the authors register the image-stacks?

Image processing software we used this time (TRI/3D-BON) didn’t have the function to register the image-stacks, therefore stacked images were reconstructed by determining a reference point so that they had the same cross section.

5. Discussion:

- Page 20, line 246. This is not correct, as this was not the aim of this paper. Please, rewrite it. Besides, do not use the word biofilm

Following your advice, we revised this part as below.

“This work assessed the influence of HFW irradiation in vivo in a rat model and investigated the target of HFW.”

- Page 22, line 271. "RCT successfully removes most bacteria leaving only that which has invaded deep inside the dentinal tubules". This is incorrect, as several studies already demonstrated that 30-60% of the root canal walls remain untouched after root canal preparation

As this reviewer suggested, it has been reported that residual bacteria in isthmus and lateral branches are difficult to remove by conventional RCT. In a rat root canal treatment model, Yoneda et al. reported that 75% of the bacteria from untreated root canals were eradicated by conventional root canal treatment and led to healing, but the remaining 25% were difficult to remove by conventional RCT only. We modified this part as below.

“Furthermore, residual bacteria decreased significantly after RCT alone, compared with the level before treatment. Yoneda et al. reported that RCT removed 75% of bacteria in the root canal and that 75% of cases were cured with conventional RCT; however, the remaining 25% of bacteria were difficult to remove [26]”

Reviewer #3: Thank you for revising your manuscript and your hardworking is appreciated by the reviewer.

Although the authors addressed most of the previous comments from the reviewer, it is still unclear in the M&M session.

For example, in each paragraph, authors used the same expression of " Twelve rats were divided equally into an irradiated group (n = 6) and a non-irradiated group (n = 6)" twice, and "The rats were divided into an irradiated group (n = 5) and a non-irradiated group (n = 5)." once.

I understand that there were 34 rats in total included in this study, and they were divided into three groups: MicroCT, Quantification of bacteria in the root canal, and Immunohistochemical observation. For each method, there were 12, 10, and 12 subjects included, respectively.

However, the M&M was still not clear and the repetitive description of "Twelve rats were divided equally into an irradiated group (n = 6) and a non-irradiated group (n = 6)" will cause confusion and may require readers to go back and forth to check the subject allocation.

Please revise this session accordingly.

Thank you very much for your comments. Following your suggestion, we revised the description about the number of rats allocated in each experiment. In materials and methods “Animals” section, allocation of all 34 rats are provided and also in each experiment, we modified the description of the number of rats used for better understanding.

Also, the sentence in Line 55 "treatment (RCT) often fails because the anatomical complexity of the root canal system, which includes lateral branches and isthmuses [2] and extraradicular biofilms [3-6], makes it difficult to completely remove the source of the bacterial infection [7]." is still inappropriate. The meaning of this sentence is inexplicit because it can be interpreted as "RCT often fails". In fact, initial NSRCT has a success rate around 95%.

Some have reported a success rate of approximately 95% for NSRCT, but this represents the success rate for pulpectomy in which no lesions have formed at the root apex, and does not represent the success rate of infected root canal treatment of teeth with pulp necrosis and apex lesion formation. The success rate of treatment of pulp-necrotic teeth with preoperative lesions is reported in the following article to be 82%.

Treatment outcome in endodontics: the Toronto study--phase 4　J endod,34(3):258-263 2008.

However, we thought it would be better to weaken the tone of the argument, so we modified that part as “Root canal treatment (RCT) can fail due to the anatomical complexity of the root canal system, which includes lateral branches and isthmuses [2], and extraradicular biofilms [3-6], which makes it difficult to completely remove the source of the bacterial infection [7].” to tone down. We appreciate your comments.

---

## [Editor Report · Decision Letter 2]

11 Sep 2020

Assessment of the functional efficacy of root canal treatment with high frequency waves in rats

PONE-D-20-08421R2

Dear Dr. Maezono,

We’re pleased to inform you that your manuscript has been judged scientifically suitable for publication and will be formally accepted for publication once it meets all outstanding technical requirements.

Kind regards,

Sompop Bencharit, DDS, MS, PhD, FACP

Academic Editor

PLOS ONE

Additional Editor Comments (optional):

The authors have sufficiently addressed all comments.
---

## [Editor Report · Acceptance letter]

18 Sep 2020

PONE-D-20-08421R2 

Assessment of the functional efficacy of root canal treatment with high-frequency waves in rats 

Dear Dr. Maezono:

I'm pleased to inform you that your manuscript has been deemed suitable for publication in PLOS ONE. Congratulations! Your manuscript is now with our production department. 

Kind regards, 

on behalf of

Dr. Sompop Bencharit 

Academic Editor

PLOS ONE